# The Assessment and the Within-Plant Variation of the Morpho-Physiological Traits and VOCs Profile in Endemic and Rare *Salvia ceratophylloides* Ard. (Lamiaceae)

**DOI:** 10.3390/plants10030474

**Published:** 2021-03-03

**Authors:** Rosa Vescio, Maria Rosa Abenavoli, Fabrizio Araniti, Carmelo Maria Musarella, Adriano Sofo, Valentina Lucia Astrid Laface, Giovanni Spampinato, Agostino Sorgonà

**Affiliations:** 1Department of Agricultural Sciences, “Mediterranea” University of Reggio Calabria, Feo di Vito, 89124 Reggio Calabria, Italy; rosa.vescio@unirc.it (R.V.); mrabenavoli@unirc.it (M.R.A.); carmelo.musarella@unirc.it (C.M.M.); vla.laface@unirc.it (V.L.A.L.); gspampinato@unirc.it (G.S.); 2Department of Agricultural and Environmental Sciences—Production, Landscape, Agroenergy, Università Degli Studi di Milano, Via G. Celoria 2, 20133 Milan, Italy; fabrizio.araniti@unirc.it; 3Department of European and Mediterranean Cultures, Architecture, Environment, Cultural Heritage (DiCEM), University of Basilicata, Via Lanera 20, 75100 Matera, Italy; adriano.sofo@unibas.it

**Keywords:** gas exchanges, leaf mass area, rare species, *Salvia ceratophylloides* Ard., VOC, within-plant plasticity

## Abstract

*Salvia ceratophylloides* (Ard.) is an endemic and rare plant species recently rediscovered as very few individuals at two different Southern Italy sites. The study of within-plant variation is fundamental to understand the plant adaptation to the local conditions, especially in rare species, and consequently to preserve plant biodiversity. Here, we reported the variation of the morpho-ecophysiological and metabolic traits between the sessile and petiolate leaf of *S. ceratophylloides* plants at two different sites for understanding the adaptation strategies for surviving in these habitats. The *S. ceratophylloides* individuals exhibited different net photosynthetic rate, maximum quantum yield, light intensity for the saturation of the photosynthetic machinery, stomatal conductance, transpiration rate, leaf area, fractal dimension, and some volatile organic compounds (VOCs) between the different leaf types. This within-plant morpho-physiological and metabolic variation was dependent on the site. These results provide empirical evidence of sharply within-plant variation of the morpho-physiological traits and VOCs profiles in *S. ceratophylloides*, explaining the adaptation to the local conditions.

## 1. Introduction

*Salvia ceratophylloides* Ard. is a perennial herbaceous species endemic to Southern Calabria, Italy, which was declared extinct in 1997 [1] and recently rediscovered a hundred mature individuals distributed in two sites, 2 Km apart, around the Reggio Calabria hills [2,3]. Nowadays, the *S. ceratophylloides* populations are threatened with extinction because of the habitat modification and destruction by wild urbanization and agriculture [3]. Spread of alien species, such as *Cascabela thevetia*, *Ipomoea setosa* subsp. *pavonii*, and *Tecoma stans*, favored by climate change, establishes an additional threat factor [4,5]. Despite its vulnerability, no quantitative information on the morphological and ecophysiological traits of *S. ceratophylloides* are available, although they are pivotal for understanding the habitat requirements for the conservation of the endemic species.

The viability of endangered and rare species, such as *S. ceratophylloides*, depends on their capability to maintain or even increase their fitness under short- and/or long-term continuous climate change. Because of its rarity, this endemic species usually pointed out specific and narrow habitat requirements suggesting that their responses must occur only in the actual habitat determining local adaptation through phenotypic plasticity and/or genetic variation [6].

Although the results sometimes appeared conflicting, the plant phenotypic plasticity assumed a significant role in the viability of rare and endangered species. For example, Noel et al. [7] observed a high degree of phenotypic plasticity that conferred an increase of the fitness in *Ranunculus nodiflorus* Ten. suggesting the maintenance of the micro-environment heterogeneity as a habitat-based strategy for its conservation. On the contrary, Westerband et al. [8] observed low phenotypic plasticity in response to drought stress in *Schiedea obovate* (Sherff) W.L. Wagner & Weller, an endangered, endemic Hawaiian shrub, showing a high risk of extinction in the future climate change scenarios. Based on earlier works of the Winn [9,10] and De Kroon’s hypothesis [11], which dealt with plant phenotypic variation at sub-individual level (i.e., organ or module), the ‘within-plant’ rather than “among-plant” phenotypic variation could represent the major source of population-level diversity in several functional traits [12,13]. Recent works pointed out the multiple ecological aspects of the ‘within-plant variation’ such as the improvement of the exploitation of the heterogeneous-distributed resource [14,15], the adaptation to biotic and biotic gradients [16], the spreading of the ecological breadth of species and individuals [17,18], the increase of the functional diversity of populations [17], and the alteration of plant-antagonist interactions [19,20,21,22]. However, no quantitative characterization of the within-plant variation in endangered and rare plant species has been assayed yet.

In this respect, we started a two-year field study (2016–2017) to characterize the *S. ceratophylloides* through a leaf-level morpho-ecophysiological and metabolic approach from its natural habitat. In particular, this species exhibited a leaf morphology characterized by contemporary presence (petiolate leaf) and the absence of petiole (sessile leaf) in the basal and upper part of the shoot, respectively, suggesting a potential within-plant variation. Furthermore, individuals of this species were in two different sites, at a distance of <2 km from each other and preliminary results by SSR, pointed out that the populations of *S. ceratophylloides* exhibited a low level of genetic variability between the two populations [23] and indicating the phenotypic plasticity is the main driver for the local adaptation. In particular, we focused on the photosynthetic performances as a marker of tolerance and growth of species to predict the optimum habitat conditions for the conservation of rare species [24] but also to provide the capacity of plant adaptation to new conditions associated with climate change and the likely changes in plant communities [25]. Further, we considered the leaf mass per area (LMA) as a morphological trait strictly correlated with the functional syndrome and consequently to plant growth and development and, ultimately, plant fitness [26]. Simultaneously, the metabolic profiles of volatile organic compounds (VOCs) could provide information on the plant status and the plant-, microbial- and arthropode-plant communications [27]. The evaluation of all these functional traits could offer useful information for effectively promoting translocation and mitigation operations to restore *Salvia ceratophylloides* rare plant species. 

In this framework, the present study was addressed to assess the morpho-ecophysiological and metabolic traits of *S. ceratophylloides* from its natural habitat and investigate the following questions: (1) does the within-plant variation of the photosynthetic performance, morphological traits, and metabolic profiles occur? (2) is the within-plant morpho-physiological and metabolic variation of *S. ceratophylloides* modified between the two localities?

## 2. Results

### 2.1. Physiological Performances and Morphological Traits of S. ceratophylloides

The net photosynthetic rate (*P*_N_) and the photosynthetic photon flux density (I) curves of *S. ceratophylloides* leaves were well-fitted (R^2^ > 0.95 and *p* < 0.05) and –described by Ye mathematical model [28] (Figure 1). The parameters estimated by non-linear regression have been reported in Table 1. As observed, the leaf type produced a significant and strong effect on most photosynthetic parameters. In particular, the sessile leaves showed a higher I_sat_ (1578 μmol(photon) m^−2^ s^−1^), I_max_ (766 μmol(photon) m^−2^ s^−1^), *P*_Nmax_ (11.22 μmol(CO_2_) m^−2^ s^−1^) and ϕ_(I_0_ – I_comp_)_ (0.030 µmol(CO_2_) µmol(photon) ^−1^) level than petiolate ones which showed a lower dark respiration rate (*R*_D_) (0.69 µmol(CO_2_) m^−2^ s^−1^). This pattern was observed in both sites except for the *P*_Nmax_, which was statistically different between the two leaf types in relation to the site (*p* < 0.05 LT × Sit interaction, Table 1): the petiolate leaves (6.85 µmol(CO_2_) m^−2^ s^−1^) showed a lower value of *P*_Nmax_ respect to the sessile ones (19.70 µmol(CO_2_) m^−2^ s^−1^) at Mo site. Further, the significant LT × Sit interaction observed for the I_comp_ and *R*_D_ indicated that the parameters difference between the leaf types was affected by the site factor. Indeed, the I_comp_ of the sessile leaf was higher than petiolate ones (8 vs. 22 µmol(photon) m^−2^ s^−1^) at Mo site only, and the same pattern was observed for the *R*_D_ (0.53 vs. 1.40 µmol(CO_2_) m^−2^ s^−1^) (Table 1). 

The comparisons of the I_sat_ and I_comp_ of *S. ceratophylloides* with those of different functional plant groups are shown in Appendix A. The minimum I_sat_ value of *S. ceratophylloides* fell between that of schlerophylls in habitat at a high light intensity and heliophytes, while the maximum one was within the range between heliophytes and C4 plants. The minimum I_comp_ value of *S. ceratophylloides* was between epiphytes and spring geophytes, while the maximum was between spring geophytes and heliophytes.

The stomatal conductance and transpiration rate of both leaves of *S. ceratophylloides*, measured at 200 and 800 µmol(photon) m^−2^ s^−1^ corresponding to the light intensities around to the I_max_ values of the P and S leaves, respectively, are reported in Table 2 and Table 3. Like photosynthetic pattern, the P leaves pointed out a significantly lower stomatal conductance and transpiration rate than S at both light intensities. However, this effect was different between the sites (*p* < 0.01 for LT × Sit interaction, Table 2 and Table 3): the S leaves showed higher levels of the stomatal conductance and transpiration rate with respect to the P ones only in the Mo site while any difference between the leaf type has been produced in Pu site. The site’s effect was highly significant (*p* < 0.001, Table 2 and Table 3) for both ecophysiological parameters, evidencing higher values for the Mo site than Pu. 

The leaf morphology of *S. ceratophylloides* was reported in Table 4. Leaf type affected the leaf fresh weight, leaf area, leaf water content, and fractal dimension, which were higher in P leaves. This pattern, however, was modified in relation to the site for the LFW and LA only with higher values in the petiolate in Pu site only (*p* < 0.05 LT × Sit interaction, Table 4). Finally, the site factor affected the LFW, LDW, and LWC showing higher values in the Mo site (Table 4). 

Value average (±SD) of LMA of *S. ceratophylloides* in comparison with that of the sun- and shade-species herbs, evergreen angiosperm, evergreen species, herbs, and different *Salvia* species are shown in Appendix A. The LMA range of *S. ceratophylloides* fell in that of the herbs, evergreen species, *S. mellifera*, *S. hispanica*, *S. officinalis*, and sun species.

### 2.2. VOCs Analysis of Salvia ceratophylloides in Its Habitat

The PCA of GC-MS spectra of different leaves and sites of *Salvia ceratophylloides* are shown in Figure 2. The two-dimensional PCA score plot revealed a separation in VOCs profile induced by leaf type, clearer in Mo site than Pu ones. The VOCs profile was also different between the two *Salvia* sites.

VOCs detected by GC-MS are summarized in Appendix A. Thirty-nine compounds belonging to different chemical classes were identified. Among them the most representative were monoterpenes (17), sesquiterpenes (7), monoterpene alcohols (4), aldehydes (4), ketons (3), alcohols (2), aliphatic esters (1) and ether (1). 

Comparing the amount of the XCMS-extracted peak intensities of each chemical between the S and P leaf types, 13 compounds emitted by both P and S were statistically different (Table 5). In particular, *p*-cymene, sabinene, terpinolene, *β*-pinene, *γ*-terpinene, *α*-terpineol, *α*-cubebene, *α*-muurolene, isovaleraldehyde, 5-methylheptan-3-one, pentan-3-one, *β*-tujone, and dimethyl sulfide were higher in S than P leaves. However, the higher emission of *β*-tujone and α-terpineol in S leaves was only observed in Mo site (significant LT × Sit interaction, Table 5). The same pattern was revealed for D-germacrene, which was not modified by both leaf type and the site as single factors. Only six compounds were differently affected by sites: *p*-cymene, α-terpineol, α-copaene and α-cubebene, emitted in Pu more than Mo site which, conversely, produced more *β*-tujone and dimethyl sulfide (Table 5).

## 3. Discussion

### 3.1. The Assessment of the Morpho-Physiological Traits of the Rare Salvia ceratophylloides Ard.

*Salvia ceratophylloides* Ard., an endemic, rare, and critically endangered plant species, has been recently rediscovered on some sites around the Reggio Calabria hills [2,3]. The knowledge of its morphological and ecophysiological traits is very important for understanding the habitat requirements for the conservation [24] and providing its capacity to adapt to climate change conditions and, consequently, to the plant communities distribution [25]. The responses of these traits in *S. ceratophylloides* are reported here for the first time. We mainly focused on two functional traits, the photosynthetic light-response curve and leaf mass per area, which indicate the habitat preferences and responsiveness to environmental conditions [24,29,30]. The photosynthetic response curves to the PAR photon flux and, in particular, the I_sat_ (818–1588 µmol(photon) m^−2^ s^−1^) and I_comp_ values (8–26 µmol(photon) m^−2^ s^−1^) suggested that *S. ceratophylloides* was well adapted to the sunny habitat. Indeed, the minimum and maximum values of the I_sat_ fell within the range defined by sclerophyll of sunny habitat and C4 plants and, the I_comp_ values were included between spring geophytes and heliophytes. The LMA values (44.1–55.4 g m^−2^) of *S. ceratophylloides* were comprised in the LMA range of the herbs [29] and evergreen species [30]. Among the different *Salvia* species, the LMA of *S. ceratophylloides* was similar to *S. mellifera*, *S. hispanica* and *S. officinalis* [31,32,33], lower than *S. mohavensis* and *S. dorrii var. dorrii* [31], and higher than *S. glutinosa* and *S. pratensis* [34,35]. The different ranges of the LMA of *S. ceratophylloides* with respect to that of some *Salvia* species, were probably correlated with its functional response to the environmental conditions, such as water and light availability [29,30]. For example, *S. mohavensis* and *S. dorrii var. dorrii* showed a higher LMA value because of the adaptation to their native desert area (mountain ranges of the Mojave Desert of southern California, south-western Nevada, and northern Baja California Norte, Mexico) [31]. Although the *S. pratensis* was strictly related to *S. ceratophylloides* (belonging to the same sect. Plethiosphace: [36]) as distributed to the similar area (native from Europe: [37]), it showed lower LMA value [34] probably due to the different growing conditions of *S. pratensis* (pot and growth chamber) in Mommer’s experiments. Finally, the LMA values (57 g m^−2^) of *S. ceratophylloides* fell in the range of sun species confirming its preference to the open sunny habitat as reported for the most *Salvia* species [32,38].

### 3.2. Does the Within-Plant Variation of the Photosynthetic Performance, Morphological Traits and Metabolic Profiles Occured? 

Recently, the within-plant variation as an expression of intraspecific phenotypic plasticity is strongly taken into account for its role in plant evolution and ecology at the individual, population, and community levels [12,13]. For example, the within-plant variation in leaf morpho-physiological traits allows the adaptation of each individual to optimize (i) its capturing structures to the heterogeneous local environmental conditions such as light, temperature, and CO_2_ gradients within plant canopy in trees [14] and perennial herbs [17], and (ii) its cost-expensive defenses against herbivory and pathogens [39,40]. Further, the knowledge of the leaf-level photosynthetic performances within the plant allow to scale at canopy level [41] and understand the competitive strategies for exploring the within-canopy heterogeneous light and CO_2_ availability. In this respect, the within-plant variation of *S. ceratophylloides* by comparing the morpho-physiological and metabolic traits of petiole and sessile leaves were here tested, for the first time. 

The *S. ceratophylloides* individuals exhibited a clear within-plant variation determined by the differences in most functional traits [I_sat_, I_max_, *P*_Nmax_, ϕ_(I_0_ – I_comp_)_, RD, stomatal conductance, transpiration rate, leaf fresh weight, leaf area, leaf water content, fractal dimension, and VOCs] between sessile and petiolate leaves. Indeed, comparing the leaf types, S pointed out a better photosynthetic performance pointing out a higher net photosynthetic rate, maximum quantum yield, and I_sat_ [1578 vs. 911 µmol(photon) m^−2^ s^−1^]. Conversely, they exhibited a lower leaf area and capacity to fill the space, as evidenced by FD, respect than the P ones.

Hence, a within-plant efficient subdivision of the functional traits for the resource acquisition and use (light, CO_2_) was observed in *S. ceratophylloides* with the short-term low-expensive physiological traits in S leaves (uppermost) and the long-term high-expensive morphological traits in P leaves (lowest). Hence, the *S. ceratophylloides* leaves, placed at the top of the shoot and facing high light intensity, temperature, and vapor pressure deficit, could use higher photon and CO_2_ fluxes for increased their carbon gains by high I_sat_ and stomatal conductance. The spatial distribution of more efficient structures or functions within plants to fit the micro-environmental heterogeneous conditions and maximize photosynthesis and carbon gain was already known in different plant species [42]. Besides the functional traits linked to the resource acquisition, the two leaf types of *S. ceratophylloides* pointed out difference in the stomatal conductance and transpiration with the S leaf showing higher values, limiting the leaf overheating, as reported by Lin et al. [43] in dry areas species. The lower aerodynamic resistance caused by the smaller size and shape of the sessile leaves (lower leaf area and fractal dimension) was further beneficially allied for avoiding leaf overheating [43,44] but also for reducing the water loss by smaller total leaf area. Overall, the leaf position in the *S. ceratophylloides* canopy showed different strategies to cope with the fine-scaled environmental gradients: from higher to lower light intensity and from dry to wet conditions along the uppermost sessile and lowest petiolate leaf gradient.

The within-plant variation of *S. ceratophylloides* was also observed in the VOCs composition and emission. Indeed, the S and P leaves were sharply separated by VOC-based metabolic profiles suggesting different intensity and composition between them. Further, the emission of 14 out of 39 identified VOCs was statistically increased in S leaves respect than P ones, including monoterpenes (p-Cymene, Sabinene, Terpinolene, β-Pinene, γ-Terpinene, and α-Terpineol), sesquiterpenes (D-Germacrene, α-Cubebene, α-Muurolene) and green leaf volatiles ((3z)-3-Hexenyl acetate) mostly involved in defenses against herbivory and pathogens [45] and in responses to abiotic stress [46]. The within-plant variation of VOCs emission in response to herbivory has been already observed in wild and crop species [47,48], but no evidence at the field level has been reported yet. Why do *S. ceratophylloides* plants defend the S more than P leaves by higher VOCs emission? Probably, the upper, younger, sessile leaves are more protected in views of their high-performing photosynthetic machinery and nutritive value (high leaf dry content, although not statistically supported) as suggested by the optimal defense hypothesis [39,40]. Overall, these results pointed out the within-plant functional subdivision at morpho-physiological and metabolic levels of *S. ceratophylloides* mimicking what has been already observed in the wide crown of trees for heat tolerance [49], light acquisition, and differential expression of genetic polymorphisms in the sun and shade leaves of trees [50] and defense responses to herbivory [51].

The within-plant variation of morpho-physiological and metabolic traits was affected by the site suggesting that *Salvia* plants were adapted to local conditions. Indeed, the S and P leaves’ morpho-physiological patterns changed between the two sites for most traits (on 12 that showed the leaf type factor as statistically significant, nine traits pointed out LT × Sit interaction). The S leaves pointed out a higher photosynthetic rate, stomatal conductance, and transpiration rate associated with higher dark respiration and I_comp_ than P ones in the Mo site only. These results explained the phenotypic plasticity response to the local conditions [52,53] because preliminary results showed a low genetic variability of *S. ceratophylloides* populations [23]. It has been observed that the within-plant phenotypic variation responded to the microhabitat environmental heterogeneity or fine-grained (small-scale) environmental variations [9,11,14,15] rather than macro-geographical or coarse-grained environmental variations [17,54,55]. In this respect, we hypothesized that the *S. ceratophylloides* individuals in the Mo site, characterized by a significant within-plant variation of the leaf morpho-physiological traits, could face with a higher microhabitat environmental heterogeneity, especially for light intensity and/or temperature gradients (the most important abiotic stresses affecting the leaf growth), than Pu site. Opedal et al. [56] observed that the microhabitat environmental heterogeneity increased with the topographically complex sites changing the intraspecific traits of 16 plant species. In accordance, the Pu site is characterized by flatter, more open terrains and higher altitude than Mo ones, which conversely is placed at a lower altitude at the base of the valley, closed and with rough terrains likely determining a short duration of light and steeply thermal and light gradients. 

Unlike the morpho-physiological traits, the within-plant variation of metabolic profiles was lesser affected by the different sites. Indeed, the PCA pointed out that S and P leaves’ metabolic profiles were separated at both sites, and only 4 out 13VOCs exhibited a statistically significant Lt × Sit interaction. Since the VOCs emission is more involved to the biotic stress (plant-plant, plant-herbivory, and plant-pathogen interactions) [27], probably the *Salvia* plants in both sites are faced to similar biotic environment heterogeneity or variability (predation, competition, etc.) differently to the abiotic ones (light, temperature, etc.), determining thus the maintaining of the same within-plant VOCs emission in *S. ceratophylloides* at both sites.

Finally, the metabolite profiling and physiological traits (10 out of 10 parameters, considering the LT × Sit interaction also; Table 1, Table 2 and Table 3, Figure 2) varied more than morphological features (4 out 7 parameters; Table 4). Why do *S. ceratophylloides* plants choose to invest in the physiological and metabolic capacity more than morphological-related traits in S and P leaves? The physiological plasticity has a low cost and more rapid response with respect to the cost-expensive morphological plasticity, allowing the expression of the most adequate plant phenotype in response to the variable climatic conditions. For example, the plant physiological plasticity is strictly related to an enhanced ability to colonize gaps and open areas and, hence, exploiting the transient environmental resources at low cost by short-term adjustments, such as plant acclimatization [47,48,49,50,51,52,53,54,55,56,57,58,59,60]. Conversely, the plant morphological plasticity is more functional for the plant adaptation in the long-term and probably useful for growing in forest understories [57,61]. The choice to invest in higher plasticity of the physiological traits than morphological ones has also been reported by Herrera et al. [17], which observed that the within-ramet variation in *Helleborus foetidus* L. was more due to the stomatal features than leaf size- and area-related traits causing an increase of seed number produced by each individual [13]. 

## 4. Materials and Methods

### 4.1. Species and Sites

*Salvia ceratophylloides* Ard. rediscovered by Crisafulli et al. [2], is a perennial herb, scapose hemicryptophyte, with woody and upright stems with a dense pubescence of glandular and simple patent hairs (Appendix A). Leaves are opposite pinnate-partite with toothed lobes and morphologically distinct in petiolate (basal, 12 × 4 cm long and less discrete pinnate lobes and presence of the petiole) and sessile (cauline, 3–4 × 1–2 cm long and more incise pinnate lobes, clasp the stem) (Appendix A). Inflorescences are 20–30 cm in length with 5–6 verticillasters, each with 4–6 flowers. 

*S. ceratophylloides* plants were identified in two small hilly and closer sites (<2 Km) around the city of Reggio Calabria, Mosorrofa (Mo), and Puzzi (Pu) (Southern Italy) (Appendix A). Each site consisted of around 60 and 240 individuals for Mosorrofa and Puzzi, respectively. Further, Mosorrofa site is topographically complex in a little valley, while Puzzi pointed out open and flatter terrains (Appendix A). Both sites are characterized by layers of loose sand alternating with benches of soft calcarenites of Pliocene origin. Soils have a sandy texture with a basic pH falling into the group Calcaric Cambisols [62].

Species identification is in agreement with Pignatti [63], and the specimens are deposited in the Herbarium of Mediterranean University of Reggio Calabria (acronym REGGIO).

The *S. ceratophylloides* location pointed out a Mediterranean climate with average annual temperatures of 18 °C and an average annual rainfall of 600 mm mostly in autumn-winter and a dry summer period. According to Rivas-Martínez [64], the macro-bioclimate is “Mediterranean pluviseasonal oceanic” (upper thermo-Mediterranean thermotype and lower subhumid ombrotype).

### 4.2. Measurements and Samplings 

Measurements and samplings have been carried out in the early summer (May–June) of 2016 and 2017 on two leaf types: (1) the upper sessile (S) (two to three nodes from the apex) and (2) the lower petiolate (P) (from fifth to sixth nodes from the apex) (Figure 1C). For the morphological and physiological analysis, 4–9 and 5–9 samples for petiolate and sessile leaves, respectively, have been collected from each site; while the metabolic analysis has been carried out on 3 leaves of each type and site. 

### 4.3. Physiological Analysis

The net photosynthetic light response curves have been determined using LI-6400XT portable photosynthesis system (Li-Cor, Inc. Lincoln, Nebraska, USA) at 2000, 1500, 800, 400, 200, 100, 30, 15, and 0 µmol m^−2^ s^−1^ irradiance levels. The net photosynthesis has been measured at 500 cm^3^ min^−1^ flow rate, 26 °C leaf temperature, 400 μmol(CO_2_) mol(air)^−1^ CO_2_ concentration (controlled by CO_2_ cylinder). Each measurement was made with a minimum and maximum wait time of 120 and 200 s, respectively, and matching the infrared gas analyzers for 50 μmol(CO_2_) mol(air)^−1^ difference in the CO_2_ concentration between the sample and reference before any change in irradiance levels. The leaf-to-air vapor pressure difference has been set at 1.5 kPa, continuously monitored around the leaf during the measurements and maintained at a constant level by manipulating incoming air humidity as needed. The measurements were carried out on sunny days between 8:30–11:30 am. Finally, stomatal conductance (*g*_s_, mol H_2_O m^−2^ s^−1^) and transpiration rate measurements (T, mmol H_2_O m^−2^ s^−1^) have been evaluated at each light intensity. 

### 4.4. Morphological Analysis

After the physiological analysis, the same leaves were used for the morphological analysis. In particular, leaf fresh weight (LFW, g) and dry weight (LDW, g), were determined after oven-drying at 70 °C for 2 days, while leaf area (LA, cm^2^) were scanned at a resolution of 300 dpi (WinRhizo STD 1600) and measured using WinRhizo Pro v. 4.0 software package (Instruments Régent Inc., Chemin Sainte-Foy, Québec, Canada). Further, the leaf fractal dimension (FD) was obtained by the “fractal analysis module” (WinRhizo software), based on the box-counting method with the following settings: maximal pixel size (2.0 mm), box sizes ranging from 2 to 32, filters, and a length/width ratio smaller than 2.00. The FD provides information on the object’s space occupation: the fractal dimension approaches the value of 2 as the leaves become dense to the point of “filling in” a shape. For this reason, the FD has been used both for correlating the root architecture to the soil resource acquisition [65], and in the LAI-Light interception models as the correction parameter [66].

Through these parameters, we also calculated the leaf mass per area (LMA, g LDW cm^−2^ LA), strongly related to photosynthetic rate [67], growth rate [68] and decomposition rate [69]; the leaf dry content (LDC, g dry weight/g fresh weight), strongly related with relative growth rate [70], flammability [71], and post-fire regeneration strategy [72]; and the leaf water content or leaf succulence (LWC, g H_2_O/cm^2^ leaf area], directly correlated with the plant responses to abiotic stresses [73]. 

### 4.5. VOC Analysis: HeadSpace/Solid-Phase Micro-Extraction (HS/SPME) GC-MS Analysis

The volatiles (VOCs) produced by petiolate and sessile leaves of *S. ceratophylloides* have been characterized using the HS/SPME method. One gram of plant material, per sample and replicate (N = 3), was sealed in a 20 mL vial and allowed to equilibrate for 20 min at room temperature. Successively, the SPME gray fiber (StableFlex, divinylbenzene/Carboxen on polydimethylsiloxane coating; 50/30 μm coating; Supleco) was exposed to plant VOCs for 20 min to allow the VOCs adsorption on the fiber.

VOCs were identified using a Thermo Fisher gas chromatography apparatus (Trace 1310) coupled with a single quadrupole mass spectrometer (ISQ LT). The capillary column was a TG-5MS 30 m × 0.25 mm × 0.25 µm. Helium was used as carrier gas with a flow of 1 mL/min. Samples were injected in a split mode with a split ratio of 60. Injector and source were settled at the temperature of 200 °C and 260 °C, respectively. The temperature ramp was settled as follow: 7 min at 45 °C, from 45 °C to 80 °C with a rate of 10 °C × min, from 80 °C to 200 °C with a rate of 20 °C × min then isocratic for 3 min 200 °C. Mass spectra were recorded in electronic impact (EI) mode at 70 eV, scanning the 45–500 *m/z* range. 

Native raw chromatograms (RAW), previously converted in mzXML using the tool MSconvert of proteowizard [74], were normalized for TIC intensity, aligned, deconvoluted, and peak intensities extracted using the open-source software XCMS [75]. For peak analysis the GC/Single Quad (matchedFilter) pre-settled method was applied. 

After chromatograms processing and peak picking, features and normalized peak areas were imported to Excel for further statistical analysis. Compounds identification was carried out comparing the relative retention time and mass spectra of molecules with those of commercial libraries (NIST Mass Spectral Reference Library) and open-source EI spectral libraries (Mass Bank of North America, Golm Metabolome Database) [76,77].

### 4.6. Statistical Analysis 

Light curves were fitted by nonlinear regression using the Ye et al. [28] model equation:(1)$$P_{N}=\phi_{\left(I_{0}-I_{\mathrm{comp}}\right)}\times \frac{1-\beta \times I}{1+\gamma \times I}\times \left(I-I_{\mathrm{comp}}\right)$$
where: *P*_N_ is the net photosynthetic rate [μmol(CO_2_) m^−2^ s^−1^]; I is the photosynthetic photon flux density [μmol(photon) m^−2^ s^−1^]; I_comp_ is the light compensation point [μmol(photon) m^−2^ s^−1^]; β is the adjusting factor (dimensionless); γ is the adjusting factor (dimensionless); ϕ_(I_0_–I_comp_)_ is the quantum yield obtained at the range between I_0_ and I_comp_ [μmol(CO_2_) μmol(photon)^−1^]. The following leaf-level photosynthetic parameters were calculated by these equations [28]:(2)$$P_{gmax}=\phi_{\left(I_{0}-I_{\mathrm{comp}}\right)}\times \frac{1-\beta \times I}{1+\gamma \times I}\times \left(I-I_{\mathrm{comp}}\right)+R_{D}$$
(3)$$R_{D}=\phi_{\left(I_{0}-I_{\mathrm{comp}}\right)}\times I_{\mathrm{comp}}$$
(4)$$I_{\mathrm{sat}}=\frac{\sqrt{\frac{\left(\beta +\gamma \right)\times \left(1+\gamma \times I_{\mathrm{comp}}\right)}{\beta -1}}}{\gamma }$$

I_sat_ is the light saturation point [μmol(photon) m^−2^ s^−1^]; *P_gmax_* is the asymptotic estimate of the maximum gross photosynthetic rate [μmol(CO_2_) m^−2^ s^−1^]; *R*_D_ is the dark respiration rate [μmol(CO_2_) m^−2^ s^−1^]. Finally, the ϕ_(I_comp_–I200)_ was calculated as the slope of the linear regression of *P*_N_ for values of I between Icomp and 200 μmol (photon) m^−2^ s^−1^ representing the “maximum quantum yield”.

Finally, according to Lobo et al. [78], we reported the Imax (μmol(photon) m^−2^ s^−1^) (light saturation point beyond which there is no significant change in *P*_N_) and the *P*_N_ (I_max_) (μmol(CO_2_) m^−2^ s^−1^) (maximum value of *P*_N_ obtained at I = I_max_) instead of I_sat_ and *P_gmax_* as more realistically adequate. We used a simple routine to minimize the error sum of squares (SSE) for fitting the models, allowing the determination of equation parameters using the Microsoft Excel spreadsheet and Solver function (Microsoft Excel 2010). Non-linear regressions were repeated several times in order to minimize the sum of square of deviation between predicted and experimental values to less than 0.01% between two consecutive fits [79].

In order to evaluate the effect of the Years (Y) (2016 and 2017), we used the one-way ANOVA on the gas exchange parameters, which quickly respond to the environmental conditions. Since the Years factor was not significant (*p* > 0.05) for almost all the gas exchange traits (Appendix A), all the morpho-physiological parameters were analyzed by two-way analysis of variance with the Leaf Type (LT) (sessile and petiolate) and Site (Sit) (Mosorrofa and Puzzi) as main factors and their interaction Lt × Sit. Then, Tukey’s test was used to compare the means of all the parameters of each LT and Sit. All the data were tested for normality (Kolmogorov–Smirnoff test) and homogeneity of variance (Levene median test) and, where required, the data were transformed.

For the comparison of the LMA and *P*_N_ of *S. ceratophylloides* with that of different plant functional groups and *Salvia* spp., we used the Isat and Icomp data obtained from [80,81] for evergreen angiosperm, [29] for evergreen shrub, [30] for evergreen species, [82] for *S. officinalis*, [34] for *S. pratensis*, [35] for *S. glutinosa*, [33] for *S. hispanica*, [31] for *S. mohavensis*, *S. leucophylla*, *S. dorrii* var. *dorrii* and *S. mellifera*.

The TIC intensity normalized dataset obtained from metabolomic data analysis were classified through unsupervised Principal Component Analysis (PCA) where the output consisted of score plots to visualize the contrast among different samples. PCA analysis was carried out on all the features detected by the analysis. Successively, identified and annotated compounds were statistically analyzed through univariate two-way analysis of variance with the LT (sessile and petiolate) and Sit (Mosorrofa and Puzzi) as major factors. Then, the Tukey’s test was used to compare the compound means of each leaf type and site (*p* < 0.05).

## 5. Conclusions

The eco-physiological adaptation of *S. ceratphylloides*, a rare and endangered plant species, to its habitat by functional traits was studied. The higher light saturation and compensation point and leaf mass per area indicated a sunny habitat preference of *S. ceratophylloides*. These results suggested that lower competition (low density and diversity), especially with woody species (trees and shrubs), should be favored for its *in situ* conservation. However, the *S. ceratophylloides* habitat has been destroyed and continuously fragmented because of anthropogenic disturbance and environmental deterioration. Consequently, further and deepening study needs to identify the main stressful factors that threaten its growth, development, and fitness. Due to the objective difficulty to preserve the taxon in situ and pending further experimental data on its ecology, as indicated for other nationally threatened CWRs species [83], *ex situ* conservation actions are recommended. For the *ex situ* propagation, we recommend growing the seedlings at least half sunlight (1200 µmol (photons) m^−2^ s^−1^). 

Further, for the first time, the “continuous within-plant variation” of the morpho-physiological traits and metabolic profiles of this species was assessed in the field. The results indicated that the physiologic and metabolic traits explained most of its within-plant plasticity, which was also affected by the location. Indeed, the sessile and petiolate leaves of *S. ceratophylloides* showed different photosynthetic performances and metabolic profiles, but the sub-individual variation of the photosynthetic-related parameters, differently to the volatilome, was exhibited in one site only. These within-plant patterns, probably related to the micro-environmental heterogeneity, could optimize the growth and defense machinery for the fitness’s improvement to specific habitats. Overall, the magnitude of the within-plant variation should be taken into consideration when designing sampling schemes for the ecological studies of *S. ceratophylloides*.

## Figures and Tables

**Figure 1 plants-10-00474-f001:**
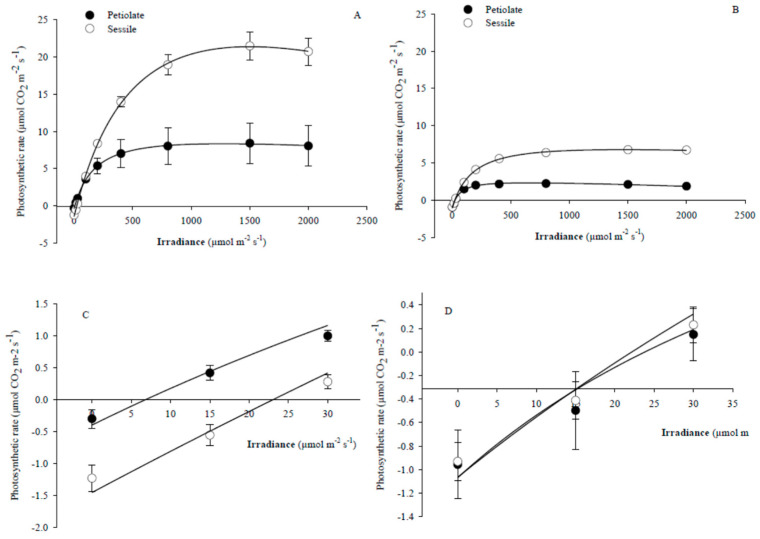
Leaf photosynthetic light-response curves measured on petiolate (●) and sessile leaves (○) of the *Salvia ceratophylloides* at (**A**,**C**) Mosorrofa site (Mo) and (**B**,**D**) Puzzi site (Pu). The (**C**,**D**) panels showed the curves at the lowest irradiance values. Data points represent means (*n* = 4–9). Light curves have been fitted by non-linear regression using the Ye et al. model [28].

**Figure 2 plants-10-00474-f002:**
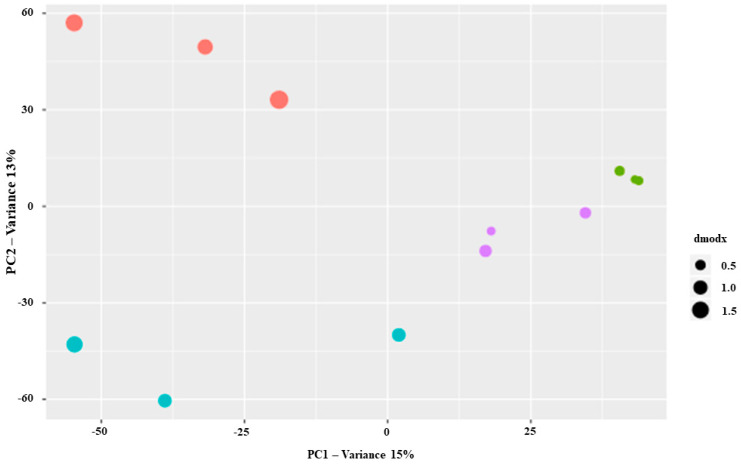
Principal component analysis of untargeted metabolomics data from different leaves (sessile and petiolate) and sites [Mosorrofa (Mo) and Puzzi (Pu)] of Salvia ceratophylloides individuals: Mo-sessile (red), Mo-petiolate (green), Pu-sessile (blue) and Pu-petiolate (purple).

**Table 1 plants-10-00474-t001:** Leaf-level photosynthetic parameters of different leaf types (P: petiolate; S: sessile) of *Salvia ceratophylloides* individuals of two sites (Mosorrofa, Mo; Puzzi, Pu) estimated by non-linear regression using the Ye et al. model [28]. Different lower-case letters represent significant differences at *p* < 0.05 among the average within the column (Tukey’s test). Different capital case letters represent statistically significant differences among the means along the rows (*p* < 0.05, Tukey’s test).

		Site (Sit)		
	Leaf Type (LT)	Mo	Pu	Leaf Type Average
I_comp_ [µmol(photon) m^−2^ s^−1^]	P	8^b^	26^a^	18^x^
	S	22^a^	22^a^	22^x^
	Site average	16^A^	23^A^	
I_max_ [µmol(photon) m^−2^ s^−1^]	P	312^b^	310^b^	311^y^
	S	839^a^	725^a^	766^x^
	Site average	655^A^	577^A^	
I_sat_ [µmol(photon) m^−2^ s^−1^]	P	1027^b^	818^b^	911^y^
	S	1559^a^	1588^a^	1578 ^x^
	Site average	1323^A^	1313^A^	
P_N(Imax)_ [µmol(CO_2_) m^−2^ s^−1^]	P	6.85^b^	2.17^b^	4.25^y^
	S	19.70^a^	6.51^b^	11.22^x^
	Site average	14.00^A^	4.96^B^	
R_D_ [µmol(CO_2_) m^−2^ s^−1^]	P	0.53^b^	1.07^ab^	0.69 ^y^
	S	1.40^a^	1.09^ab^	1.19 ^x^
	Site average	1.01^A^	1.08^A^	
ϕ_(I_0_-comp)_ [µmol(CO_2_) µmol(photon) ^−1^]	P	0.025^b^	0.0095^c^	0.016^y^
	S	0.046^a^	0.021^b^	0.030^x^
	Site average	0.037^A^	0.016^B^	

**Table 2 plants-10-00474-t002:** Leaf-level stomatal conductance and transpiration rate of different leaf types (P: petiolate; S: sessile) of *Salvia ceratophylloides* individuals of two sites (Mosorrofa, Mo; Puzzi, Pu) measured at a light intensity of 200 µmol m^−2^ s^−1^. Different lower-case letters represent significant differences at*p*< 0.05 among the average within the column (Tukey’s test). Different capital case letters represent statistically significant differences among the means along the rows (*p* < 0.05, Tukey’s test).

		Sites (Sit)		
	Leaf Type (LT)	Mo	Pu	Leaf Type Average
Stomatal conductance (mol H_2_O m^−2^ s^−1^)	P	0.032^b^	0.016^b^	0.023^y^
	S	0.113^a^	0.029^b^	0.059^x^
	Site average	0.077^A^	0.025^B^	
Transpiration rate (mol H_2_O m^−2^ s^−1^)	P	0.87^b^	0.55^b^	0.69^y^
	S	2.59^a^	0.92^b^	1.52^x^
	Site average	1.82^A^	0.79^B^	

**Table 3 plants-10-00474-t003:** Leaf-level stomatal conductance and transpiration rate of different leaf types (P: petiolate; S: sessile) of *Salvia ceratophylloides* individuals of two sites (Mosorrofa, Mo; Puzzi, Pu) measured at a light intensity of 800 µmol m^−2^ s^−1^. Different lower-case letters represent significant differences at*p*< 0.05 among the average within the column (Tukey’s test). Different capital case letters represent statistically significant differences among the means along the rows (*p* < 0.05, Tukey’s test).

		Sites (Sit)		
	Leaf Type (LT)	Mo	Pu	Leaf Type Average
Stomatal conductance (mol H_2_O m^−2^ s^−1^)	P	0.032^b^	0.016^b^	0.023^y^
	S	0.107^a^	0.032^b^	0.059^x^
	Site average	0.074^A^	0.026^B^	
Transpiration rate (mol H_2_O m^−2^ s^−1^)	P	0.83^b^	0.55^b^	0.67^y^
	S	2.44^a^	1.03^b^	1.53^x^
	Site average	1.72^A^	0.86^B^	

**Table 4 plants-10-00474-t004:** Biometric and morphological parameters of different leaf types (P: petiolate; S: sessile) of *Salvia ceratophylloides* individuals of two sites (Mosorrofa, Mo; Puzzi, Pu). Different lower-case letters indicated significant differences at*p*< 0.05 among the average within the column (Tukey’s test). Different capital case letters indicated statistically significant differences among the means along the rows (*p* < 0.05, Tukey’s test).

		Sites (Sit)		
	Leaf Type (LT)	Mo	Pu	Leaf Type Average
Leaf fresh weight [g leaf^−1^]	P	1.33^a^	1.38^a^	1.36^x^
	S	1.40^a^	0.43^b^	0.78^y^
	Site average	1.37^A^	0.77^B^	
Leaf dry weight [g leaf^−^]	P	0.23^a^	0.22^b^	0.22^x^
	S	0.27^a^	0.12^b^	0.17^x^
	Site average	0.25^A^	0.16^B^	
Leaf area [cm^2^]	P	41.4^ab^	54.4^a^	48.6^x^
	S	43.6^ab^	19.9^b^	28.3^y^
	Site average	32.2^A^	42.6^A^	
Leaf mass x area [g m^−2^]	P	55.4^a^	44.1^a^	49.1^x^
	S	62.3^a^	61.6^a^	61.8^x^
	Site average	59.2^A^	55.3^A^	
Leaf dry content [g dry weight g^−1^ fresh weight]	P	0.17^a^	0.18^a^	0.18^x^
	S	0.19^a^	0.28^a^	0.25^x^
	Site average	0.18^A^	0.24^A^	
Leaf water content [g H_2_O cm^−2^ leaf area]	P	0.027^a^	0.020^a^	0.023^x^
	S	0.025^b^	0.016^b^	0.019^y^
	Site average	0.026^A^	0.017^B^	
Fractal dimension	P	1.67^a^	1.73^a^	1.70^x^
	S	1.51^b^	1.65^b^	1.56^y^
	Site average	1.69^A^	1.59^A^	

**Table 5 plants-10-00474-t005:** Chemical characterization of volatile organic compounds in fresh sessile and petiolate leaves of two sites [Mosorrofa (Mo) and Puzzi (Pu)] of Salvia ceratophylloides plants.

	Compound	^#^ Statistics	Sessile		Petiolate	
			Pu	Mo	Pu	Mo
1	p-Cymene	LT 7.78 * Sit 14.16 ** LT × Sit 0.21 ^NS^	195,813	82,392	108,569	19,607
2	Pinocarvone	LT 0.04 ^NS^ Sit 6.57 * LT × Sit 0.07 ^NS^	2406	987	2444	696
3	Sabinene	LT 11.80 ** Sit 0.34 ^NS^ LT × Sit 1.70 ^NS^	554,775	873,306	195,242	73,210
4	Terpinolene	LT 12.40 ** Sit 0.40 ^NS^ LT × Sit 3.09 ^NS^	128,554	218,320	62,198	19,784
5	β-Pinene	LT 7.30 * Sit 0.47 ^NS^ LT × Sit 1.73 ^NS^	92,968	150,052	53,391	35,502
6	γ-Terpinene	LT 5.40 * Sit 0.19 ^NS^ LT × Sit 0.04 ^NS^	16,341	13,366	4610	3508
7	α-Terpineol	LT 8.13 * Sit 12.91 ** LT × Sit 9.04 *	10,220^b^	80,003^a^	11,854^b^	18,054^b^
8	D-Germacrene	LT 0.11 ^NS^ Sit 3.47 ^NS^ LT × Sit 4.22 *	2554^a^	169^b^	1102^a^	1218^a^
9	α-Copaene	LT 0.62 ^NS^ Sit 11.05 * LT × Sit 0.67 ^NS^	3517	601	2385	625
10	α-Cubebene	LT 8.21 * Sit 19.35 ** LT × Sit 1.02	4,460,201	1,371,705	2,247,264	312,465
11	α-Muurolene	LT 9.49 * Sit 0.56 ^NS^ LT × Sit 0.99 ^NS^	14,382	15,236	7105	1038
12	Isovaleraldehyde	LT 6.10 * Sit 0.52 ^NS^ LT × Sit 0.52 ^NS^	81,876,770	46,391,341	3,426,789	3,466,464
13	5-Methylheptan-3-one	LT 5.70 * Sit 0.21 ^NS^ LT × Sit 0.08 ^NS^	7776	8204	1578	3291
14	Pentan-3-one	LT 7.73 * Sit 2.44 ^NS^ LT × Sit 1.20 ^NS^	321,989	649,080	114,170	171,753
15	β-tujone	LT 17.37 ** Sit 6.21 * LT × Sit 12.54 **	65,370^b^	168,599^a^	54,660^b^	36,692^b^
16	(3z)-3-Hexenyl acetate	LT 3.58 ^NS^ Sit 5.09 ^NS^ LT × Sit 5.46 *	1253^a^	0^b^	99^ab^	122^ab^
17	Dimethyl Sulfide	LT 23.77 ** Sit 5.34 * LT × Sit 1.40 ^NS^	29,866,633	54,386,837	3,926,181	11,857,751

# Statistical analysis: two-way ANOVA with 4–9 replications (LT: leaf type; Sit: sites; LT × Sit: Leaf type × Sites interaction); * 0.05 >*p*< 0.01; ** 0.01 >*p*< 0.001; NS: not significant; Different letters along the row indicated significant differences among the means (*p* < 0.05, Tukey’s test).

## Data Availability

The data presented in this study are available on request from the corresponding author.

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
