# Peer review of "The Assessment and the Within-Plant Variation of the Morpho-Physiological Traits and VOCs Profile in Endemic and Rare Salvia ceratophylloides Ard. (Lamiaceae)"

_plants, 2021, doi:10.3390/plants10030474_

Round 1

Reviewer 1 Report

The authors propose a manuscript titled “The assessment and the within-plant variation of the morpho-physiological traits and VOCs profile in endemic and rare Salvia ceratophylloides Ard. (Lamiaceae)”

The article is deep in the evaluations and well structured. In particular the study takes into consideration an endemic and rare species: Salvia ceratophylloides, and analyze the adaptation at local conditions in order to preserve itself. Tne authors discuss also about the variation of the morpho-ecophysiological and metabolic traits between the sessile and petiolate leaf in two different sites. Minor revision are required for its publication.

Introduction

I suggest to add a distribution map of this species. The authors in the text give an indication of the sites, but for the reader who reads the article in an international context, could not know where is located Calabria and in particular the species sites.

Rows 34-35. Please add this local reference on the species discussed

Bonsignore, C.P.; Laface, V.L.A.; Vono, G.; Marullo, R.; Musarella, C.M.; Spampinato, G. Threats Posed to the Rediscovered and Rare Salvia ceratophylloides Ard. (Lamiaceae) by Borer and Seed Feeder Insect Species. Diversity 2021, 13, 33. Doi: 10.3390/d13010033

Conclusion

Rows 567-569 Please modify and complete the sentence in this way:

development, and fitness. Due to the objective difficulty to preserve the taxon in situ and pending further experimental data on its ecology, as indicated for other nationally threatened CWRs species (Perrino and Wagensommer 2021), ex situ conservation actions are raccomanded. For the ex situ propagation, we recommend growing the seedlings at least half sunlight….”.

Please add a reference to complete the period

Perrino, E. V.; Wagensommer, R. P. Crop wild relatives (CWR) priority in Italy: distribution, ecology, in situ and ex situ conservation and expected actions. Sustainability 2021, 13, 1682. https://doi.org/10.3390/su13041682

Author Response

1) The article is deep in the evaluations and well structured

Thanks a lot

2) I suggest to add a distribution map of this species. The authors in the text give an indication of the sites, but for the reader who reads the article in an international context, could not know where is located Calabria and in particular the species sites.

I added a new map.

3) Rows 34-35. Please add this local reference on the species discussed

done

4) Rows 567-569 Please modify and complete the sentence in this way: “development, and fitness. Due to the objective difficulty to preserve the taxon in situ and pending further experimental data on its ecology, as indicated for other nationally threatened CWRs species (Perrino and Wagensommer 2021), ex situ conservation actions are recommended. For the ex situ propagation, we recommend growing the seedlings at least half sunlight….”. Please add a reference to complete the period: Perrino, E. V.; Wagensommer, R. P. Crop wild relatives (CWR) priority in Italy: distribution, ecology, in situ and ex situ conservation and expected actions. Sustainability 2021, 13, 1682. https://doi.org/10.3390/su13041682

Done

Reviewer 2 Report

The study is well-conducted and centered on different leaf morphology and site.  Very well done and complete.  My major remark is that the fine language must be improved to present a more readable product.  To this end, I would suggest that the authors have a competent native speaker or editing service evaluate and make suggestion. I did not edit the entire manuscript.

Two primary points:

  1. Move the first paragraph of the M and M to the introduction.  As I was reading your ms. I was wondering how the two different morphological leaf variants were placed on the plant.  This would go a long way in improving the disposition of the reader.
  2.  For the data/information retrieved from other publications, do you need permission from either the authors/publishers or both.  Might be a good idea to check on this.

Ln 71  Genetic diversity -- see comment on paper.  You need to be much more descriptive and detailed about this.  An abstract is not an appropriate citation as they are generally not available and insufficient details are presented.

I have included an annotated file for your consideration.

Nice Manuscript and very complete.

Author Response

1) The study is well-conducted and centered on different leaf morphology and site. Very well done and complete.

Thanks a lot

2) My major remark is that the fine language must be improved to present a more readable product. To this end, I would suggest that the authors have a competent native speaker or editing service evaluate and make suggestion. I did not edit the entire manuscript.

Done. Checked by native speaker.

3) Move the first paragraph of the M and M to the introduction. As I was reading your ms. I was wondering how the two different morphological leaf variants were placed on the plant. This would go a long way in improving the disposition of the reader.

I think that description of the plant material should be in M&M as the description of the other materials. However, if you think that the best option is to move in the Introduction, I'll move there.

4) For the data/information retrieved from other publications, do you need permission from either the authors/publishers or both. Might be a good idea to check on this.

I checked. I created a new figure from data showed in tables and figures of the other papers. Redrawn figure using data or results from other publications, do not require copyright permission. Further, I cited the sources.

5) Ln 71 Genetic diversity -- see comment on paper. You need to be much more descriptive and detailed about this. An abstract is not an appropriate citation as they are generally not available and insufficient details are presented.

I added more information.

6) I have included an annotated file for your consideration.

I answered directly on the pdf file that upload in the new versions.

7) Nice Manuscript and very complete.

Thanks a lot
